# Exogenous Antioxidants in Remyelination and Skeletal Muscle Recovery

**DOI:** 10.3390/biomedicines10102557

**Published:** 2022-10-13

**Authors:** Ricardo Julián Cabezas Perez, Marco Fidel Ávila Rodríguez, Doris Haydee Rosero Salazar

**Affiliations:** 1School of Medicine, Faculty of Health, Universidad Antonio Nariño, Bogota 110111, Colombia; 2Department of Clinical Sciences, Universidad del Tolima, Ibagué 730007, Colombia; 3School of Dentistry, University of Washington, Seattle, WA 98105, USA

**Keywords:** antioxidants, inflammation, demyelination, nerve, skeletal muscle, regeneration, remyelination, myelin, oxidative stress

## Abstract

Inflammatory, oxidative, and autoimmune responses cause severe damage to the nervous system inducing loss of myelin layers or demyelination. Even though demyelination is not considered a direct cause of skeletal muscle disease there is extensive damage in skeletal muscles following demyelination and impaired innervation. In vitro and in vivo evidence using exogenous antioxidants in models of demyelination is showing improvements in myelin formation alongside skeletal muscle recovery. For instance, exogenous antioxidants such as EGCG stimulate nerve structure maintenance, activation of glial cells, and reduction of oxidative stress. Consequently, this evidence is also showing structural and functional recovery of impaired skeletal muscles due to demyelination. Exogenous antioxidants mostly target inflammatory pathways and stimulate remyelinating mechanisms that seem to induce skeletal muscle regeneration. Therefore, the aim of this review is to describe recent evidence related to the molecular mechanisms in nerve and skeletal muscle regeneration induced by exogenous antioxidants. This will be relevant to identifying further targets to improve treatments of neuromuscular demyelinating diseases.

## 1. Introduction

Inflammation and tissue disorder are outcomes of the imbalance in the production of reactive oxygen species (ROS) accompanied by insufficient activity of endogenous antioxidants that boost oxidative stress [1,2]. The increase of ROS such as superoxide and hydroxyl radicals may cause DNA lesions, protein structural damage, oxidize lipids, and mitochondrial dysfunction ending in apoptosis [3,4,5,6]. These events are associated with neurodegenerative and demyelinating disorders including Alzheimer’s disease, Parkinson’s disease, and multiple sclerosis [1]. In addition, low levels of exogenous antioxidants exacerbate ROS effects and subsequent degenerative effects [1,2,5].

Myelin is a sphingolipid-based multilayer produced by oligodendrocytes and Schwann cells [7,8]. This multilayer by wrapping the nerve axons forms the isolation to improve neuronal action potential maintaining the speed of nerve impulses [7,8]. Autoimmune and chronic inflammatory diseases in the nervous system exhibit removal of the myelin sheath, or demyelination, in central and peripheral nerves [8,9]. Demyelinating diseases outline a wide group of clinical conditions with heterogeneous clinical outcomes [9]. These outcomes include severe nerve injury resulting in myopathy-like damage, increased collagen production, or fibrosis, and loss of muscle fibers [10,11,12]. Early mechanisms related to these events were observed in rat models of nerve injuries showing massive ROS production in the oxidative stage [11]. This occurs along with increased protein kinases activity and hypoxia-inducible factor-1 (HIF-1) during cellular adaptation as compensatory mechanisms [11]. All this shows that nerve damage and demyelination impair skeletal muscle arrangement.

Demyelination is not described as a direct cause of myopathies, however, there is extensive muscle damage derived from impaired innervation as occurs in multiple sclerosis and other demyelinating diseases [13]. This is similar to what is observed in autoimmune myopathies as dermatomyositis with severe inflammation and comparable to the damage observed in chronic inflammatory demyelinating polyneuropathy [11,14,15]. The neuromuscular junctions also show impairment due to the oxidative and inflammatory reactions in nerves and muscle fibers [13,14]. Hence, autoimmune disorders either of the nervous system or skeletal muscle result in inflammation, increased oxidative stress, demyelination, and skeletal muscle damage.

The endogenous antioxidant system regulates oxidative balance by reducing ROS damage through specific mechanisms such as ROS scavenging and restriction of ROS generation [16]. These mechanisms neutralize ROS production by binding metal ions and decreasing oxidative effects [16]. Antioxidant molecules include superoxide dismutase (SOD), catalase, peroxidase, and glutathione, all able to detoxify ROS and keep balanced aerobic events [16]. Endogenous antioxidants seem to promote remyelination in vitro and in vivo for what stimulating production and activity through exogenous mechanisms might be promising in nerve and skeletal muscle regeneration [5,17,18,19]. Exogenous antioxidants such as polyphenols and others are showing protective effects in nerve and skeletal muscle cells by reducing free radicals [20]. Further research on the mechanisms of exogenous antioxidants in recovering demyelinated nerves and derived impaired skeletal muscles will feature targets to stimulate remyelination. Therefore, the aim of this review is to provide an overview of the recent evidence on the molecular mechanisms in nerve and skeletal muscle regeneration promoted by exogenous antioxidants. This will be relevant to identifying adjuvant targets to enhance the treatments of inflammatory and autoimmune neuromuscular demyelinating diseases.

## 2. Autoimmune and Inflammatory Demyelination

Myelination is the process performed by the specialized cells oligodendrocytes and Schwann cells to produce sphingolipids and form myelin layers [7]. The sphingolipids produced by oligodendrocytes including sphingomyelin, cerebrosides, and gangliosides, exhibit key roles in proliferation, migration, apoptosis, and remyelination [7,21]. Hence, failed sphingolipid synthesis may induce pathological and inflammatory reactions.

### 2.1. Role of Sphingolipids and ROS

The brain has the highest sphingolipid content and therefore requires a crucial balance in the synthesis, degradation, and removal of lipids and sphingolipids [22,23,24]. Changes in the lipid levels may trigger pathogenic reactions including neuroinflammation and oxidative damage [22,23,24,25]. For instance, defective sphingolipid metabolism may cause neurodegenerative diseases such as Alzheimer’s and Parkinson’s disease [22,23,24,25]. In a study including 30 patients diagnosed with multiple sclerosis was observed increased oxidative damage, lipid alteration, and DNA lesion [25]. In addition, these patients showed plaque formation containing oligodendrocytes, immunoproteasome activity, and astrocytes similar to brain biopsies from previous cases [21,25]. It was also observed inflammatory CD3-positive T-cells and oxidative stress markers in blood and cerebrospinal fluid including high concentrations of malondialdehyde and lipid hydroperoxide [6,25]. All this indicates oxidative effects and neuroinflammation associated with sphingolipid alteration and ROS production.

Also, oligodendrocytes under in vitro inflammatory conditions fail to differentiate and undergo apoptosis [21]. In vivo, a mice model for multiple sclerosis showed a reduction in oligodendrocyte proliferation, increased immune cell activation, immunoproteasome activity, increased demyelination, and decreased remyelination [21]. Immunoproteasome pathways are observed in both oligodendrocytes and astrocytes surrounded by a chronic inflammatory environment [21,26]. Interestingly, these pathways in astrocytes induce protective effects against oxidation and damage, while in oligodendrocytes accelerate apoptosis and tissue injury [21,26]. The immunoproteasome activity is one of the molecular mechanisms involved in demyelination and impaired remyelination yet to be understood. 

The crosstalk between the sphingolipid-mediated pathway and the eicosanoid pathway may also impair myelination [7,27]. The sphingolipid-mediated pathway is associated with tissue development while the eicosanoid pathway is linked to pathological events such as the production of inflammatory mediators [7,27]. Orm1-like3 (ORMDL3) is an endoplasmic reticulum protein involved in leukotrienes synthesis regulation and sphingolipid homeostasis [28]. In vitro, mast cells isolated from wild-type mice and knockout-ORMDL3 mice showed higher production of sphingolipids and inflammatory mediators such as leukotrienes [28]. Low ORMDL3 expression induces failed sphingolipids production along with inflammatory reactions by leukotrienes [28]. In vivo, ORMDL1, ORMDL2, and ORMDL3 knockout mice showed altered sphingolipid synthesis during myelination resulting in severe demyelination in the sciatic nerve [29]. It seems that ORMDLs might be target proteins crucial for the stability in sphingolipid production and myelination.

Similarly, ceramide, ceramide1-phosphate (C1P), and sphingosine1-phosphate (S1P) may induce inflammatory reactions [7,30]. These intermediates are synthesized in the sphingomyelin cycle by sphingomyelinases [7,31]. Some studies show the critical balance of sphingolipids to regulate apoptotic or anti-apoptotic signals [21,22]. For instance, increasing ceramide levels may reduce S1P levels activating cell death pathways [21,22]. In vitro and in vivo using models of multiple sclerosis and inflammatory demyelinating diseases show that ceramides accumulate and activate astrocytes [30]. Sphingosine increases while S1P decreases, and both may be reestablished once remyelination occurs [30]. Ceramide, C1P, and S1P participate in the activation of protein kinase C isoforms and cytosolic phospholipase A (cPLA) [7,22,27]. This latter enzyme is associated with inflammation via the arachidonic acid pathway and activation of EDG-sphingolipid receptors. This leads to the activation of phospholipase C (PLC) and intracellular calcium release (Figure 1) [7,22,27].

Additionally, it is considered a synergistic action of C1P with S1P that induces COX-2 and activates PLA2 [7,21,32]. C1P contributes to inflammation via the production of prostaglandin E2 (PGE2) and stimulates mast cell degranulation via Ca^+^2 dependent pathway associated with calmodulin [7,21,32,33]. Similarly, in the activated mast cells the sphingosine kinase produces S1P via the IgE receptor [7,21,32]. These events also occur in different cell types such as lung fibroblasts and epithelial cells. S1P stimulation via tumoral necrosis factor (TNF) induces COX-2 expression and triggers PGE2 production ending up in severe inflammation [7,21,32,34]. In a few words, the therapeutic regulation of the synergistic effects of C1P-S1P may reduce inflammatory reactions in other tissues preventing additional clinical conditions.

Furthermore, a metabolic syndrome involving insulin resistance is associated with a high risk of peripheral neuropathy affecting small unmyelinated axons described in recent reviews [35,36]. In brief, the neuropathy associated with type 2 diabetes includes an increase of long chain fatty acids, neuroinflammation, and neuronal oxidative stress [35]. These mechanisms induce a cascade of pro-inflammatory cytokines along with NF-kB activation and further production of COX-2 and TNF-α [35]. Interestingly, type 2 diabetes neuropathy shares COX-2 mechanisms with C1P and S1P lipid inductors [35]. Cytokine cascade induced by metabolic syndrome activates neutral SMase and SK1 with the concomitant activation of PLA2, PLC, S1P, and COX-2 mediated inflammatory effects [35,36]. Later, peripheral innervation is impaired with subsequent skeletal muscle damage.

Therefore, sphingolipids upon certain conditions prompt inflammatory responses including TNF-α induction of sphingomyelinase associated with increased ceramide levels and activation of the NF-κB pathway [21,37,38]. This induces more than 150 genes increasing cytokine and chemokine levels [21,37,38]. NF-κB activation encodes the production of interleukin-1β, IL-6, IL-8, and pro-inflammatory enzymes, such as COX-2 [21,37,38]. In murine models, ceramides stimulate in astrocytes the expression of proinflammatory mediators (TNF, IL-1B, and IL-6) and activation of NF-κB [7,22,27]. Interestingly, ceramides also increase the production of the c/EBP factor associated with the induction of TNF, IL-6, and IL-1β similar to NF-κB [7,21,32,34]. Thus, sphingolipid pathways represent potential targets to prevent inflammation, demyelination, and induce remyelination.

All this evidence suggests that sphingolipid derivatives including ceramide, C1P, and S1P, are associated with inflammatory mechanisms in the nervous system [7,22,23]. Ceramide is a key regulator of critical neuronal mechanisms, including cell differentiation, senescence, and cell death [7,22,23]. The increased levels of ceramide may induce cell death via caspase 3 activation, reactive oxygen species production, and mitochondrial dysfunction [7,22,27]. Sphingolipid metabolites constitute key mediators during inflammatory reactions being associated with neurodegenerative effects [37,38,39]. Neuroinflammatory events in neurodegenerative diseases include activation of both microglia and astrocytes showing overexpression of inflammatory mediators [37,39,40,41]. Microglia and astrocytes continuously produce pro-inflammatory cytokines conducting to chronic inflammation and the progression of neurodegeneration linked with neuroinflammation [37,39,40,41]. Such mechanisms are primary the pathophysiological foundations of neurodegenerative diseases.

### 2.2. Inflammation in Demyelinating Diseases

Overall, autoimmune and inflammatory diseases such as multiple sclerosis show increased ROS production rates [42]. Preclinical encephalomyelitis models using rodents showed higher levels of ROS produced by microglia and infiltrated macrophages [25,43]. This stimulates increasing pro-inflammatory mediators and oxidizing radicals, including superoxide, hydroxyl radicals, hydrogen peroxide, and nitric oxide causing oxidative damage [25,43]. Simultaneously, depletion of endogenous antioxidants such as catalase, SOD, and glutathione is also observed in neuroinflammation and neurodegeneration [16,44]. SOD converts superoxide radical anions into hydrogen peroxide in two forms of this enzyme: MnSOD and Cu/ZnSOD both expressed in the brain [16,44]. Nuclear factor E2-related factor (Nrf2-ARE) induces expression of glutathione and thioredoxin-antioxidant system along with peroxiredoxins, superoxide dismutase, catalases, and heme oxygenase upon inflammatory reactions and neurodegeneration [44]. Nrf2 deficiency causes in mice loss of oligodendrocyte, demyelination, neuroinflammation, and axonal damage [45,46]. Similarly, peripheral nerves from Nrf2 knockout mice exhibited reduced axonal remyelination, functional impairment, and damaged neuromuscular junctions [47].

Sensory and motor nerves are susceptible to degeneration and immune-mediated demyelination as seen in multiple sclerosis, chronic inflammatory demyelinating polyneuropathy, and Guillain-Barre syndrome [9,22,31,40]. Multiple sclerosis constitutes one of the major demyelinating diseases with effects on skeletal muscles [9,38]. Demyelination in the central nervous system is a complex process involving blood-brain barrier disturbance, immune cell recruitment, B-lymphocytes and T-Lymphocytes activation, secretion of pro-inflammatory mediators, activation of microglia, and macrophage stimulation [23,31,38]. Altogether, destabilizes structural myelin proteins including basic myelin-protein, myelin oligodendrocyte glycoprotein, myelin acidic protein, contactin1, and neurofascin [38]. The activity of the innate and adaptive immune system produces reactive oxygen species and nitrogen reactive species causing myelin damage and oligodendrocytes decreasing. These mechanisms also involve macrophage stimulation, IgM antibody-mediated damage against myelin, and demyelination [38,48,49].

Demyelination in multiple sclerosis is the final phase of the disease showing myelin sheath degradation by immune reactive cells [23,31,41]. Different types of lipids as phospholipids and sphingomyelin form the myelin sheath [23,31,41]. Myelin is also comprised of immunoglobulin-nature proteins such as myelin-associated glycoproteins (MAG), myelin oligodendrocyte glycoprotein (MOG), and cell adhesion proteins [23,31,41]. These components may induce the activation of autoimmune or immune-derived responses that causes myelin degradation [21,31,41]. First evidence shows a direct antibody-antigen interaction in multiple sclerosis in preclinical autoimmune encephalomyelitis models against MOG [21,31,41]. These models showed oxidative damage and plaque formation that, to some extent, are caused by reduced arterial perfusion and low tissue oxygen supply [23,38,41]. Then, demyelination follows the increased oxidative damage and plaques deteriorating the tissue area as seen in multiple sclerosis.

On the other hand, diseases such as neuromyelitis optica show significant immunoglobulin deposition and characteristic immunoreactivity against astrocytic aquaporin AQP4 [23,27,41]. This demyelination in the optic nerve may cover the brain stem and rarely, the brain tissue [23,27,41]. Histopathological findings show loss of axons, low AQP4 and AQP1, and the production of glial acidic fibrillary protein GFAP, a key landmark of glial reactivity [23,41]. Eosinophiles and granulocytes infiltrate the lesion causing an increased inflammatory process with the ongoing activation of macrophages and lymphocytes [23,38].

Acute disseminated encephalomyelitis (ADEM) is generally preceded by an infection or rarely by vaccination and it is associated with an immunoreactive process against MOG antigens [23,38,41]. ADEM usually affects children and young adults, and some variants are multiphasic (MDEM) [23,32,38]. The differentiation of such conditions with multiple sclerosis represents clinical challenges. ADEM and MDEM show macrophages reactivity, foamy macrophages that increase demyelination throughout the central nervous system or are restricted to a single location [23,32,38]. MOG seems to play a key role as a surface receptor and adhesion molecule as seen in demyelinating encephalomyelitis models [22,23,38]. MOG may be used as a diagnostic marker and possible target in diseases including ADEM, MDEM, and rarely multiple sclerosis in which IgG1 antibody production against MOG is found [22,23,38].

In peripheral nerves, Schwann cells produce myelin and keep a balanced nerve environment [34]. Interestingly, Schwann cells are capable to sense damaging signals and they may be involved in developing the inflammatory process via the expression of macrophage colony-stimulating factor (CSF-1) [34]. CSF-1 upregulates the production of TNF-α, and several cytokines such as IL-1α and IL-1β, to stimulate the recruitment of macrophages [34]. Trias et al. demonstrated in an amyotrophic lateral sclerosis model that Schwann cells might change their phenotype to induce nerve repair [50,51]. This change promotes proliferation and secretion of pro-inflammatory factors including CSF-1 and IL-34 from distal to proximal, as Wallerian degeneration, increasing nerve injury [50,51]. Schwann cells also remove myelin debris, ensure nerve homeostasis, and promote axonal regeneration and reinnervation [50,51]. Therefore, Schwann cells are critical and double-edge cellular components to regulate neuro-regeneration or neurodegeneration.

In demyelinating disorders involving Schwann cells, it is observed oxidative stress that impairs their mitochondrial metabolism [33,52]. For example, Friedreich’s ataxia disease occurs with loss of dorsal root ganglion seen in clinical studies along with oxidative stress that diminishes the expression of frataxin [33,52]. Frataxin seems to regulate iron molecules for proper protein functioning. The precise mechanism and downregulation of frataxin are not yet well understood. Studies in knockout models showed increasing selective inflammatory toxicity when Schwann cells are devoid of frataxin [33,52]. In fact, the absence of frataxin in Schwann cells may turn the phenotype from normal to inflammatory [33,52]. This shows a suitable target to promote peripheral nerve regeneration.

Similarly, other evidence suggests a role of Schwann cells in the balanced expression of TNF-α to promote axonal regeneration [53]. Kato et al. demonstrated via the inhibition of TNFα using etanercept-a TNFα-antagonist, improvement in axonal regeneration after nerve injury [53]. The adaptive immune response uses a major histocompatibility-II (MHC-II) complex to coordinate the T-cell activation [54]. MHC-II is expressed by antigen-presenting cells (APC) including B cells, dendritic cells, and macrophages [54]. However, inflammatory conditions may also induce the non-immune cells as endothelial or muscle cells to upregulate the expression of MHC-II [54]. Hartlehnert et al. showed that Schwann cells may upregulate MHC-II under inflammatory conditions after injury in models of neuropathic pain [54]. Intriguingly, Schwann cells may function as conditional antigen-presenting cells under inflammatory disorders [54]. All this evidence shows the role of Schwann cells in the regulation of inflammatory and immune responses following nerve damage.

To sum up, molecular mechanisms in demyelination acting as potential targets for nerve regeneration seem to be ORMDL proteins, immune proteasomes showing protective activity in astrocytes, destruction of oligodendrocytes, regulation of ceramides, sphingolipid pathways and endogenous antioxidants. Molecular events in peripheral nerves include the regulation of TNF-α, frataxin, and the ability of Schwann cells to change their phenotype turning into inflammatory. All these molecular mechanisms might be involved in skeletal muscle damage and would certainly be relevant targets to induce remyelination and nerve regeneration.

## 3. Skeletal Muscle Damage and Oxidative Stress Demyelination Related

Striated skeletal muscle tissue is formed by aligned fibers associated with the intramuscular extracellular matrix arranged into endomysium, perimysium, and epimysium [55]. The endomysium is the connective tissue surrounding each muscle fiber containing capillaries and nerve terminals [55,56,57,58]. The perimysium is a stronger connective tissue that arranges muscle fibers into small and large myobundles [55,56,57,58]. This arrangement provides mechanical stability and supports muscle contraction and force production [55,56,57,58]. The epimysium surrounds the entire muscle being related to tendons and fasciae [55,56,57,58]. The regenerative ability of skeletal muscles relies on their satellite cells, the quiescent stem cells located in the proximity of the muscle fibers [59,60,61]. Upon injury, these satellite cells activate, proliferate, fuse, and differentiate into multinucleated fibers to replace or repair the injured ones [59,60,61]. However, multiple conditions trigger the destruction of muscle fibers, defeat their regenerative capacity, and induce loss of muscle arrangement [59,62,63]. Severe conditions such as acquired or inherited diseases, demyelination and trauma exhibit such effects decreasing the muscle mass [59,62,63,64]. All this hampers the regenerative ability of skeletal muscles, increases the connective tissue, and forms fibrosis ending up in functional impairment [59,62,63].

Inherited myopathies such as dystrophies are caused by genetic factors while acquired myopathies including infectious and inflammatory diseases are linked to systemic factors [62,63]. These types of myopathies show that ROS triggering skeletal muscle damage are SOD, hydrogen peroxide, hydroxyl radical, neuronal nitric oxide, endothelial nitric oxide, and peroxynitrite [3]. Hydroxyl radical is a highly reactive oxidant that causes DNA and protein damage as occur in impaired nerves [3,11]. ROS in nerve injury induce inflammatory stages that in turn induces proteolysis in muscle fibers causing skeletal muscle impairment [11]. This damage induces the activation of TNF, TGF-β, and JAK-STAT, among other inflammatory molecules [11]. Simultaneously, decreasing of proteins involved in the regulation of oxidative stress such as Nrf2, may cause myopathy and dysfunction as recently observed in knockout mice [65]. These events induce structural changes in the intramuscular extracellular matrix causing reduction of the muscle mass and atrophy [11,66]. The permanent loss of muscle mass causes long-term functional impairment altogether defined as volumetric muscle loss [67].

Volumetric muscle loss also impairs neuromuscular junctions and induces secondary denervation as shown in rat models [67]. This secondary denervation results in chronic functional impairment [67]. In mice, passive rehabilitation after a massive muscle loss induces adaptation and minor functional recovery [68]. In general, this incomplete muscle recovery indicates persistent damage for what stimulating nerve regeneration alongside neuromuscular junction and muscle fiber may conduct an overall recovery.

As mentioned earlier in this review, the role of immunoproteasomes in nerve impairment is not well understood. In skeletal muscles, proteasomes seem to have key roles in recovery after exercise, minor injuries, and muscle atrophy [69]. In vitro, proteasomes 20S and 26S in C2C12 myoblasts and human myoblasts increase during muscle cell differentiation [69]. The inhibition of these proteasomes causes protein oxidation and stimulates proapoptotic pathways [69]. Similarly, the proteasome mechanism in atrophy is related to the regulation of the ubiquitin-proteasome system by transcription factors including Nrf1 and Nrf2 [70,71,72]. In a model of oculopharyngeal muscular dystrophy using Drosophila the increased activity of the ubiquitin-proteasome system conducts to muscle damage and impaired function [71]. The gene therapy to stimulate the production of inhibitors improved muscle weakness and reduced degeneration [71]. In other clinical conditions causing skeletal muscle atrophy, there was also increased activity of the ubiquitin-proteasome system and muscle protein degradation [72]. Intriguingly, proteasome activity seems to have effects on myoblast differentiation and muscle atrophy.

Sphingolipids and ceramides also have a role in muscle inflammation and degeneration. Recent in vitro findings showed that the impairment of ceramide kinase might be one of the molecular mechanisms causing these effects [73]. In a different in vitro study, high levels of TNF-α increased ceramides synthesis, harmed protein metabolism and induced sphingolipid accumulation impairing myofiber formation [74]. This might be related to a damaged endoplasmic reticulum involved in protein and lipid synthesis [75]. In vitro, lipotoxicity and abnormal protein folding induce the production of long-chain ceramide signals, metabolic disease, inflammation, and skeletal muscle degeneration [75]. To our knowledge, the role of ORMDL-proteins in skeletal muscle damage has not been yet described. Surely, there might be mechanisms related to ORMDL-proteins occurring during muscle degeneration as seen in nerve impairment.

In summary, molecular mechanisms inducing demyelination seem to increase skeletal muscle damage. All of this represents targets to improve regeneration in nerves and muscles. In the next section, we will discuss the potential effects of exogenous antioxidants on specific molecular mechanisms to improve remyelination and muscle recovery.

## 4. Skeletal Muscle Regeneration along Antioxidant Induced Remyelination

Remyelination involves oligodendrocytes and Schwann cell differentiation for nerve recovery [76]. In rats and mice, the inhibition of sphingomyelinase 2 (SMase2) restored myelin production and improved ceramide content in the remyelinated nerves [77]. In mice, remyelination occurs after contusion injury with the activation of the myelin regulatory factor (Myrf) [78]. Myrf produced by oligodendrocytes and Schwann cells induce myelin recovery in the impaired nerves [78]. Interestingly, Schwann cells in Myrf knockout mice continue myelin production showing expression of specific markers such as myelin protein zero (P0) [78]. Myelination by Schwann cells allowed partial recovery of the injured nerves [78]. This indicates the critical adaptation of specialized myelinating cells to sustain myelination upon injury.

Exogenous antioxidants are expected to activate molecular and anti-inflammatory mechanisms to prevent degeneration and/or induce regeneration. Thus, therapeutical antioxidant options and further research ought to target oxidative imbalance and altered cellular pathways that cause neurodegeneration and skeletal muscle damage (Figure 2). In this section, specific mechanisms of exogenous antioxidants observed in nerve and skeletal muscle will be discussed.

Major endogenous antioxidants in nerves and skeletal muscles are superoxide dismutases (SOD1, SOD2, SOD3), catalase, and glutathione [3,16,44]. Along with exogenous antioxidants such as curcumin, EGCG, vitamins, and others, endogenous antioxidants may induce regeneration by increasing protection against peroxidation [3,16,44]. Similarly, the potential effects of polyphenols such as EGCG and other natural exogenous antioxidants exhibit ROS reduction, decreasing inflammation, and myofiber formation [19,79]. The effects of exogenous antioxidants on nerve regeneration and muscle recovery are summarized in Table 1.

### 4.1. Curcumin

Curcumin is a phenolic compound with antibacterial, anti-inflammatory, and antioxidant properties with potential effects on the recovery of neurodegenerative disorders [19]. In vitro, oligodendrocyte progenitor cells treated with curcumin showed improvement in mitochondrial activity, and increased cell differentiation via PPAR-γ activation, ERK1/2 phosphorylation, and increased PGC1-α expression [80]. In vivo, a sciatic nerve crushed injury model in rats was treated using curcumin [81]. This curcumin was administered through a subcutaneously implanted pump at the site of injury [81]. There was functional and morphological muscle recovery along with reduction of oxidative markers and increased Nrf2 antioxidant in the treated groups [81]. This model shows promising results, and it may consider delivering curcumin using a less invasive strategy. In general, Nrf2 seems to be relevant in muscle recovery and nerve regeneration by reducing demyelination via exogenous antioxidant stimulation.

### 4.2. Flavonoids

Quercetin is a flavonoid found in multiple vegetables showing protective mitochondrial mechanisms, antiinflammation, and neuroprotection [19]. The flavonoid-derived medication Baicalin was recently analyzed in vitro using oligodendrocytes and in vivo in mice models of demyelination [82]. In vitro, Baicalin induced oligodendrocyte proliferation, differentiation, and a reduction in the number of astrocytes [82]. In vivo, the anti-inflammatory effect of this medication inhibited demyelination, promoted remyelination, and enhanced coordinated movement [82]. Additional neuroprotective effects of Baicalin and similar flavonoids include mechanisms to reduce neurotoxicity caused by aminochromes [83]. An aminochrome is a quinone formed during dopamine oxidation known to induce mitochondrial dysfunction and subsequent neuroinflammation in Parkinson’s disease [83]. It seems that flavonoids stimulate neuroprotection by preventing lysosomal dysfunction and protecting against oxidative damage, even though all these aspects are still under research [83]. Similarly, in a mice model of cuprizone, a model of toxic demyelination, the treatment with the flavonoid Icariin increased myelin restoration, APC^+^/Olig2^+^ mature oligodendrocytes, and brain-derived neurotrophic factor production [84]. In general, flavonoids seem to induce neuroprotection by reducing neurotoxicity showing possible therapeutical approaches, for instance, to improve treatments of Parkinson’s disease.

### 4.3. EGCG

Polyphenols such as epigallocatechin-3-gallate (EGCG) derived from green tea are showing potential regenerative effects. In vitro, C2C12 myoblasts exposed to polyphenolic EGCG showed a higher number of myotubes with increasing length [85,86]. EGCG activates the transcriptional coactivator TAZ that in turn increases myogenin, myoblasts fusion, and myotubes formation [85]. TAZ is related to tissue homeostasis, regeneration, organ development, and myogenic differentiation that occurs through the stimulation of the myogenic differentiation factor (MyoD) [85,87]. Similarly, EGCG stimulates miRNA-486-5p expression and reduces myostatin promoting C2C12 myoblasts differentiation [86]. Myostatin is a member of the transcription growth factor β (TGFβ) family that inhibits satellite cell proliferation, myoblasts differentiation, and in vivo induces fibrosis in severely injured muscles [88,89].

In a study using senescent mice was provided an enriched-EGCG diet for up to eight weeks [86]. Their results showed a gradual increasing of muscle mass in this group than in the group without this diet [86]. A similar result was found in a mice model of dystrophy that showed improvements in force production in the treated mice for up to eight weeks with EGCG than in the untreated group [90]. In the same study, the dystrophic diaphragm and soleus showed a higher area of skeletal muscle and lower area of connective tissue in time [90]. Aged rats were used in a hindlimb muscle atrophy disuse model [91]. The treatment with EGCG induced reduction of pro-apoptotic pathways, satellite cell proliferation, cell differentiation, and force production in the treated group [91]. In this group, oxidative markers were lower while SOD antioxidant markers were higher [91]. In a different study, the crushed nerve injury in the hindlimbs of rats was treated with EGCG intraperitoneally for up to 8 weeks [92]. There were increasing myelin sheath thickness, nociceptive recovery, hindlimb reflex, and posture improvements such as standing on the injured limb in the treated groups [92]. This study did not describe the functional or histological analysis of hindlimb skeletal muscles. However, the recovery in posture and coordinated movements indicate potential recovery in the injured muscles. All this evidence suggests the activation of regenerative mechanisms in nerves and skeletal muscles induced by exogenous antioxidant effects.

Another molecular mechanism involved in skeletal muscle regeneration induced by EGCG is similar to the effects observed in insulin-like growth factor (IGF-1). In vitro, IGF-1 enhances myotube formation in aligned scaffolds showing increasing length and higher number of nuclei per myotube [93]. In vivo, IGF-1 delivered in aligned scaffolds implanted in large excisional wounds showed myofiber formation and lower areas of collagen in time [93]. In recent in vitro studies, skeletal muscle fibers isolated from mice were stimulated to express Forehead box (Foxo)-O1 and Foxo-O3 [94,95]. These proteins are implicated in muscle atrophy mainly by increasing protein degradation via E3 ubiquitin ligase expression [94,95]. A decreased nuclear Foxo expression was observed in muscle fibers exposed to IGF-1 and EGCG but not in the untreated cells [95]. The addition of ROS such as H_2_O_2_ to the EGCG group cells did not increase nuclear Foxo activity [95]. All this indicates that EGCG might promote additional protective effects in skeletal muscle fibers comparable to those that occur upon IGF-1 stimulation.

In situ, in a model of severe nerve and muscle radiation injury in the rat hindlimb was delivered EGCG in a biodegradable synthetic hydrogel scaffold [96]. It was observed three months later increased antioxidant markers Nrf2 and MnSOD alongside reduced nNOS and TNF-α [96]. The area of the regenerated nerve was higher in the EGCG-hydrogel group than in the untreated group. The myelin sheath thickness and the number and diameter of axons were also higher in the treated groups with EGCG-hydrogel [96]. In the same study, the area of muscle fibers in the fibrotic muscles due to radiation exposition showed higher recovery in the EGCG-hydrogel group than in the untreated group. Similarly, strength and muscle mass were higher in the treated group [96]. The in vitro component in this study included the effects of the scaffold EGCG-loaded using Schwann cell and skeletal muscle fibers isolated from rats [96]. Their results one week after culture showed higher proliferative ability and differentiation of both cell types in EGCG-hydrogel than in control cultures [96]. All this preclinical evidence indicates that EGCG stimulates nerve regeneration and potential recovery of atrophic muscles. All this is promising for therapeutic approaches in nerve and skeletal muscle regeneration.

In a clinical study, 60 years-old patients were treated with EGCG for 12 weeks [97]. Their main results include increases in antioxidant activity, reduction of myostatin, and grip strength [97]. The autoimmune and inflammatory effects seen in multiple sclerosis disease are mostly demyelination and skeletal muscle damage [98]. A daily dose of 600mg of EGCG was provided for up to 12 weeks to patients upon treatment for multiple sclerosis [98]. After moderate exercise, improvements in muscle metabolism such as lactate reduction, and stable carbohydrate oxidation were found over time [98]. Further analysis beyond this study may include specific neural and muscular parameters for regeneration in these patients receiving EGCG along with their treatment. This clinical evidence shows regenerative effects of EGCG in the recovery of severely impaired skeletal muscle due to demyelination. In summary, EGCG seems to stimulate molecular mechanisms for satellite cell differentiation, skeletal muscle growth, and regeneration.

### 4.4. Vitamins and Other Elements

Regarding vitamins and other remyelinating components, vitamin E promotes myelin maintenance and prevents cell damage as shown in mice [99]. The lack of this vitamin causes axonopathy and demyelination [99]. The administration of vitamin E and vitamin D3 in a rat model of multiple sclerosis showed antioxidant effects, reduced apoptosis, and increased remyelination [100]. The treatment of the sciatic nerve injury in rats with vitamin E mixed with pyrroloquinoline-quinone, a small molecule with antioxidant effects, showed improvement in nerve function, increased muscle mass, and motor functional recovery [101]. Vitamin C, or ascorbic acid, is also involved in peripheral nerve development and maintenance for collagen synthesis and lipid protection showing antioxidant effects [17].

Selenium is a trace element considered an essential micronutrient for eukaryotes and procaryotes through the function of selenoproteins (Se1P). Knockout mice for this protein showed problems in learning during training and impaired motor coordination [102]. Se1P incorporates selenium in the form of selenocysteine and selenium-methionine residues [103]. In the CNS, selenium acts as a cofactor for glutathione peroxidase types I–IV and VI [104].

**Table 1 biomedicines-10-02557-t001:** Exogenous antioxidant effects on muscle and nerve regeneration.

Compound	Type	Effect on Muscle	Effect on Brain/Neurons	References
Vitamin C	Ascorbic acid	Antioxidant	Peripheral nerve development Antioxidant	[17]
Quercetin	Flavonoid	Recovery of neuromuscular function. Reduction in skeletal muscle atrophy.	Antiinflammatory neuroprotection	[19]
Baicalin	Flavone glicoside	Reduce skeletal muscle damage	Oligodendrocyte proliferation, differentiation. Improve remyelination, activation of PGC1a	[82,83]
Icariin	Flavonoid	Angiogenesis, tendon bone healing.	Myelin restoration, oligodendrocyte maturation, BDNF increase.	[84]
EGCG	Polyphenol	Increase myotubes number and length Myogenic differentiation Increase muscle area Increase antioxidative production. Reduction lactate and stable carbohydrate oxidation	Increase myelin sheath thickness Nociceptive recovery Improvements in posture	[85,86,87,88,89,90,91,92,93,94,95,96,97,98]
Vitamin E/	Tocopherol/	Antioxidant Muscle recovery	Antioxidant effects reduced apoptosis, increase remyelination. Improvements in nerve and motor function.	[99,100,101]
Vitamin D3	Cholecalciferol		Antioxidant effects Reduced apoptosis, increase remyelination	
Selenium			Antioxidant, microglial inhibition, increased remyelination	[102,103,104,105,106]

Selenium is also important in the function of SelP, a selenium-rich protein, present in glial cells, the choroid plexus, and cerebral spinal fluid [104]. SELENOP1 is a metal binding protein with antioxidant functions able to regulate the concentration of tau phosphorylated protein and synaptic Zn^2^ in a mice model [105]. Finally, 18β-glycyrrhetinic acid is a compound found in licorice roots shown to suppress the proinflammatory chemokines CCL2, CCL3, CCL5, CXCL10, and CCL20 in mice models of encephalomyelitis [106]. This study showed microglial inhibition and increased remyelination possibly via the expression of brain-derived neurotrophic factor (BDNF) [106]. All these vitamins and elements are showing potential remyelinating effects worthy of deeper research. In general, the evidence in this review is showing promising effects in remyelination and muscle regeneration mostly induced by exogenous antioxidants.

Further strategies in sync with pharmacological treatment and antioxidant approaches include physical activity. Exercise is another therapeutical intervention to reduce oxidative mechanisms following demyelination to recover muscle mass and function [107,108]. The genetic approach in mice models also aims to stimulate and accelerate myelin production reducing inflammatory responses [64]. One of the mechanisms of accelerated remyelination is the activation of peroxisome proliferator-activated receptor gamma co-activator 1-alpha (PGC1a) that increases during exercise in mice [109]. It seems that accelerating remyelination after autoimmune inflammation protects the axon structure and prevents axonal loss in the central nervous system.

To last, a growing worldwide disorder triggering neurodegeneration is obesity [35,110]. This condition increases nerve damage and shows impaired nerve regeneration also in absence of diabetes [35,110]. In obese rats, the recovery of mechanical injuries of the sciatic nerve showed a lower number of myelinated axons and thinner myelin sheath than the normal weight group [111]. Similarly, hyperlipidemia, or high levels of lipids in the blood, constitutes a risk that may induce neuropathy and peripheral nerve dysfunction [112]. Lowering weight and preventing hyperlipidemia may reduce the risk and peripheral neuropathy effects [113]. Obesity may also cause muscle damage and impair skeletal muscle regeneration [114]. Satellite cells from limb muscles in obese mice showed reduced activity of AMP-activated protein kinaseα [114]. This enzyme stimulates myogenin expression and myoblasts fusion for its inhibition decreases the regenerative ability in skeletal muscles [114]. Oxidative stress and exacerbation of multiple inflammatory responses in obesity induce the inhibition of AMP-activated protein kinaseα [114,115]. Then, based on the evidence in this review, exogenous antioxidants such as flavonoids, EGCG, and vitamin E may reduce oxidative damage and inflammation improving the regenerative ability of nerves and skeletal muscles.

To conclude, autoimmune and inflammatory mechanisms cause demyelination and nerve damage that harms skeletal muscles causing myopathy-like destruction. Oligodendrocytes and Schwann cells show adaptive responses upon injury and incomplete remyelination. Approaching mechanisms such as Nrf2, ORMDL, proteasomes, and others, might improve remyelinating therapies. Additional targets with potential pharmacological interest include the metabolic regulation of ceramides and their association with inflammatory mechanisms. Preclinical evidence using antioxidants such as curcumin, EGCG, and other exogenous antioxidants showed regenerative effects. These effects include stimulation of oligodendrocyte and Schwann cell differentiation, increased myelin production, reduction of oxidative stress, increased endogenous antioxidant activity, recovery of the neuromuscular junctions, and skeletal muscle functional recovery. Then, inducing simultaneous regeneration of nerves, skeletal muscle, and neuromuscular junctions along with current approaches will enhance the overall recovery discussed in this review.

## Figures and Tables

**Figure 1 biomedicines-10-02557-f001:**
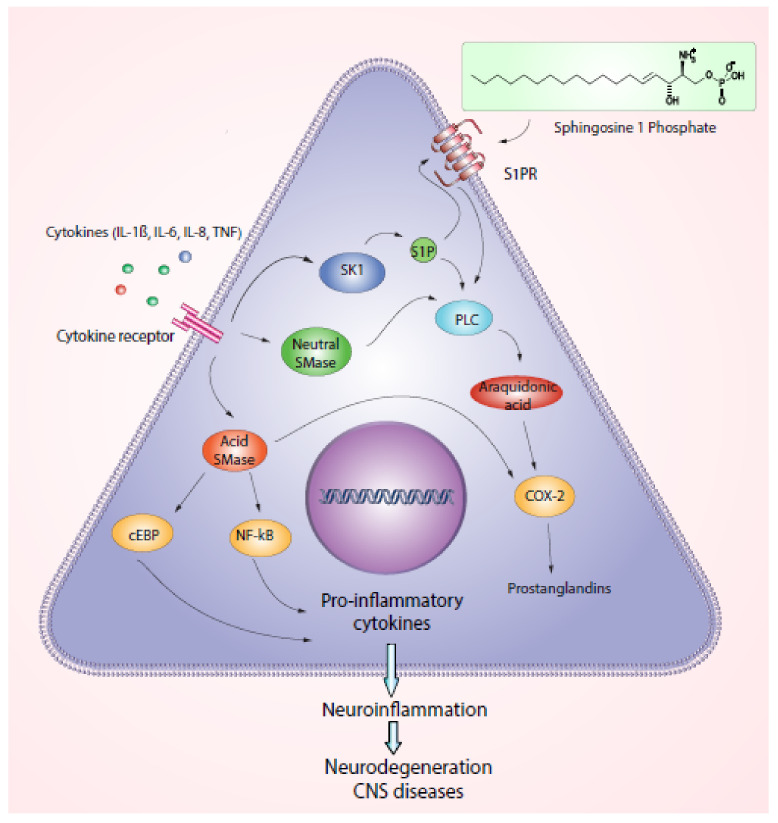
Mechanism of neuroinflammation associated with sphingolipids. Oligodendrocyte. Sphingolipids as sphingosine 1 phosphate cause the transduction of the PLC pathway with the ongoing activation of the araquidonic acid pathway. Araquinodate induces the activation of COX-2-producing prostaglandins that act as inflammatory mediators. Enzymes such as Acid sphingomyelinase SMase and Neutral sphingomyelinase are also capable of active araquidonic, and COX-2 paths. Interestingly, extracellular inflammatory mediators, including cytokines (IL-6, IL-1β, IL-8, and TNFα) may activate sphingomyelinase enzymes, mainly, acid sphingomyelinase with the subsequent activation of cEBP and NF-kB trans-activators that lead the production of pro-inflammatory cytokines. The inflammatory mediators and pro-inflammatory cytokines are associated with the development of neuroinflammation, demyelination, and degenerative diseases.

**Figure 2 biomedicines-10-02557-f002:**
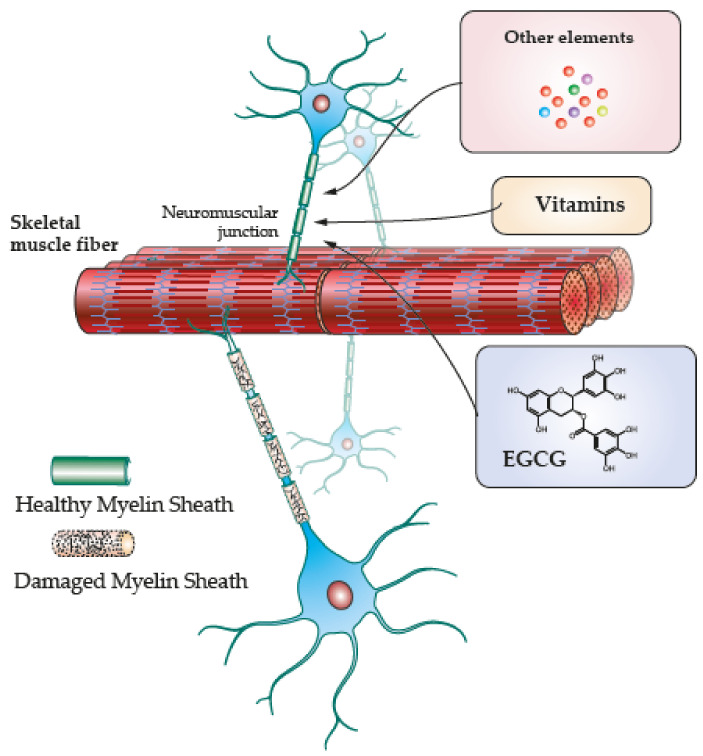
Exogenous antioxidants in regeneration. Damaged myelin impairs innervation and neuromuscular junctions end in skeletal muscle fibers damage. Remyelination is one of the potential effects of exogenous antioxidants that activate mechanisms to induce regenerative effects in skeletal muscles including their functional recovery.

## Data Availability

Not applicable.

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
