# Peer review of "Exogenous Antioxidants in Remyelination and Skeletal Muscle Recovery"

_biomedicines, 2022, doi:10.3390/biomedicines10102557_

Round 1

Reviewer 1 Report

Dear Authors,

Please, be aware with the following issues.

There are some latin expressions in the manuscript that should be written in italics. Such as, in vivo, in vitro, in situ.

Line 205. the word "physiopathological" should be substituted by the word "pathophysiological".

Lines 326-327. Improve the redaction of sentence ended ... larger groups. It appears something lack.

An abbreviations section would improve the paper and help readers to follow the message on a better way. Consider to add this new section to the manuscript. 

Reviewer 2 Report

The manuscript is interesting and has merit, however it should be further improved. I have several comments:

  1. In vivo and in vitro should be written in italics.
  2. Line 152: is it possible that PLA2 induction could also affect the skeletal muscle insuline resistance and could the latter affect the regeneration capacity? (doi/10.14814/phy2.14662)
  3. Since myelin loss is also noted in Parkinsons's diseases (doi.org/10.1371/journal.pone.0163774) and large amounts of L-DOPA act as a substantial source of ROS and aminochrome (doi.org/10.3389/fnmol.2018.00467), could mechanisms that protect from aminochrome also be protective in case of demyelinization? Please discuss briefly .
  4. Obesity and metabolic disorders are very prevalent in developed society. They can cause oxidative stress and promote inflammation and could modulate the regeneration capacity (doi.org/10.3390/ijms23020847), and could antioxidants have some specific benefits in this setting. Please discuss briefly.

Round 2

Reviewer 2 Report

The authors satisfactorily addressed all my comments.